# Niclosamide as a Promising Therapeutic Player in Human Cancer and Other Diseases

**DOI:** 10.3390/ijms232416116

**Published:** 2022-12-17

**Authors:** Zhan Wang, Junyi Ren, Jinxiu Du, Huan Wang, Jia Liu, Guiling Wang

**Affiliations:** Key Laboratory of Cell Biology, Department of Cell Biology, Ministry of Public Health and Key Laboratory of Medical Cell Biology, Ministry of Education, China Medical University, Shenyang 110122, China

**Keywords:** niclosamide, FDA-approved drug, pharmacological activity, therapeutic prospect, cancer, disease

## Abstract

Niclosamide is an FDA-approved anthelmintic drug for the treatment of parasitic infections. However, over the past few years, increasing evidence has shown that niclosamide could treat diseases beyond parasitic diseases, which include metabolic diseases, immune system diseases, bacterial and viral infections, asthma, arterial constriction, myopia, and cancer. Therefore, we systematically reviewed the pharmacological activities and therapeutic prospects of niclosamide in human disease and cancer and summarized the related molecular mechanisms and signaling pathways, indicating that niclosamide is a promising therapeutic player in various human diseases, including cancer.

## 1. Introduction

Niclosamide (NIC; 5-chloro-salicyl-(2-chloro-4-nitro) anilide) was originally developed and marketed as a molluscicide by Bayer in the late 1950s that could effectively kill snails, cercariae, trematodes, etc., and was listed as Bayluscide in 1959 [1]. In 1960, scientists at Bayer discovered that it was effective against human tapeworms, and it was marketed as Yomesan for humans in 1962 [2]. In 1982, NIC was approved by the US Food and Drug Administration (FDA) for the treatment of tapeworm infections and listed as an essential drug by the World Health Organization (WHO), though it is no longer commercially available in the United States [3]. Among the underlying mechanisms associated with the drug actions of NIC are preventing ATP synthesis by inhibiting the cellular mitochondrial oxidation and phosphorylation processes and uncoupling the electron transmission chain, as well as the modulation of Wnt/β-catenin, mTORC1, STAT3, NF-κB, Notch, and other signaling pathways. NIC is cytotoxic, but it was reported that NIC, as an oral drug, is absorbed only partially by the intestine and the absorbed portion is rapidly eliminated by the kidney; therefore, it has no cumulative toxic effect on humans [4]. In addition, NIC is not just an anthelminthic drug for treating the infection of parasites but may have broad applications for diseases beyond parasitic diseases [5]. Since NIC was approved by the FDA, it is one of the attractive candidates for drug repurposing and deserves further clinical study [6]. Recently, some studies modified NIC to obtain derivatives with higher water solubility and lower toxicity [7]. Therefore, we systematically reviewed the multifunctional pharmacological activities and therapeutic prospects of NIC in various diseases and its derivatives of the salt form, summarizing the related molecular mechanisms and signaling pathways, and indicating that NIC is a promising therapeutic player in various human diseases, including cancers.

## 2. NIC and Its Salt Forms

A reliable LC-MS/MS method for NIC detection was successfully used to perform pharmacokinetic studies in rats and dogs. NIC presented dose-independent pharmacokinetics in the dose range of 0.3–3 mg/kg after intravenous administration, and the drug exposure in rats and dogs after oral administration was very low. Additionally, NIC presented high plasma protein binding (>99.8%) and low metabolic stability [8]. A liquid chromatography–tandem mass spectrometry assay was undertaken to investigate the pharmacokinetics of NIC, which was linear from 31.25–2000 ng/mL (high dynamic range) and 0.78–100 ng/mL (low dynamic range) [9]. Meanwhile, increasing the bioavailability of NIC by blocking its metabolism by P450 enzymes will unlikely be fruitful. In contrast, the inhibition of NIC glucuronidation in both the liver and intestine may prove effective for increasing NIC’s bioavailability, thereby making it practical to repurpose this drug for treating systemic diseases [10]. Furthermore, hydroxyethyl cellulose can also enhance the solubility/dissolution of NIC [11].

NIC, which is an aniline salicylic acid with structural features associated with pleiotropic pharmacological activity [12], has two major salt forms: NIC alcohol amine salt (NEN) and NIC piperazine (NPP) (Figure 1). Due to the same efficacy and good safety, the high solubility of these two salt derivatives has attracted much attention [13].

### 2.1. NIC Ethanolamine Salt (NEN)

NEN is the ethanolamine salt form, which has higher utility than NIC because it can provide another proton [14]. At the same time, NEN is more widely used because of its low toxicity and high solubility in water [15]. Many studies showed that NEN can treat cancer, mainly digestive system cancer [16].

The mechanism of NEN in cancer mainly involves the uncoupling of oxidative phosphorylation [17], as well as the regulation of Wnt/β-catenin [18], mTORC1 [19,20], STAT3, NF-κB, and Notch signaling pathways [21], which is the same as NIC. At the same time, the application of NEN in metabolic disorders also received extensive attention, mainly in the treatment of diabetes [22,23]. The mechanism of NEN in metabolic disorders mainly involves the induction of AMPK-mediated phosphorylation of p62 (S351), leading to atypical Nrf2 activation. Nrf2 is an essential transcription factor for the elimination of lipotoxic-induced ROS [24]; therefore, Nrf2 activators are recognized as promising therapeutic targets for nonalcoholic steatohepatitis (NASH). Some studies also reported that NEN protects mouse liver from acute lipotoxic stress by activating the p62-dependent Keap1–Nrf2 pathway [25].

### 2.2. NIC Piperazine (NPP)

NIC piperazine (NPP) is another form of NIC with known safety. The water solubility of NPP is higher than that of NIC but lower than that of NEN. High water solubility usually results in higher oral bioavailability and greater potency. However, increased bioavailability may also enhance biotoxicity. Therefore, in the clinical treatment of NPP, it is most important to find a balance point between high therapeutic effects and low biotoxicity. NPP attracted significant attention in the treatment of metabolic disorders, mainly in type 2 diabetes mellitus (T2D) [26], where the main mechanism is also the mitochondrial uncoupling effect.

## 3. The Multifaceted Pharmaceutical Activities of NIC

Over the past few years, increasing evidence has shown that NIC displays multifunctional pharmaceutical activities and therapeutic prospects (Figure 2), including anti-cancer activity, metabolic regulatory activity [27], immunotherapeutic activity, and anti-viral and anti-bacterial activities [28], which can regulate multiple signaling pathways and biological processes [29], such as the Wnt/β-catenin [30], mTOR [31], STAT3 [32], NF-κB [33], and Notch signaling pathways [34]. 

### 3.1. The Anti-Tumor Activity of NIC

In 2009, Minyong Chen found that NIC serves as a negative modulator of Wnt/Frizzled1 signaling by depleting upstream signaling molecules (i.e., Frizzled and Dishevelled), which implies that NIC may be used as a tool compound to modulate Wnt/Frizzled function in the study of cancer and regeneration at the molecular level [35]. Many studies reported that NIC and its ethanolamine salt (NEN) have anti-tumor activity in many cancers (Figure 3), such as colorectal cancer (CRC), breast cancer, lung cancer, and prostate cancer, by regulating the proliferation, migration, invasion, and apoptosis of tumor cells [36] (Table 1). In addition to the anti-cancer effect of NIC and NEN alone, there are many reports that they can eliminate the chemotherapy resistance of broad-spectrum anti-cancer agents, thus improving the efficiency of chemotherapy [37]. Meanwhile, NIC can make cancer cells sensitive to immunotherapy. Here we discuss the anti-tumor activity of NIC and its ethanolamine salt, which provides a new theoretical basis for the treatment of cancer.

#### 3.1.1. NIC Inhibits Tumor Cell Proliferation

Infinite proliferation is one of the basic characteristics of tumor cells, and there are many signaling pathways involved in tumor cell proliferation. As a potential anti-cancer drug, NIC can inhibit cancer cell proliferation in a variety of cancers, such as leukemia, nasopharyngeal cancer (NPC), CRC, liver cancer, prostate cancer, and breast cancer [30]. In acute myeloid leukemia (AML), NIC inhibits the proliferation of AML cells by inhibiting the Wnt/β-catenin signaling pathway and downregulating the expression of phosphorylated CREB [38]. In chronic myeloid leukemia (CML), NIC inhibits the proliferation of CML cells by downregulating the expression of the signaling molecules STAT5 and Akt [39]. In NPC, NIC inhibits cell proliferation by downregulating ku70/80 expression, thereby increasing the radiotherapy sensitivity of NPC cells [40]. NIC inhibits the proliferation of hepatoma cells by inhibiting STAT3 signaling, thereby increasing the chemosensitivity of hepatocellular carcinoma (HCC) cells [41]. In CRC, by downregulating LEF1-mediated DCLK1 expression, NIC inhibits CRC cell proliferation and its cancer cell stemness [42]. In addition, NEN causes an inhibited proliferation of colon cancer cells by promoting mitochondrial decoupling, thus promoting pyruvate flow into the mitochondria [16,25,43]. In prostate cancer, NIC suppresses the proliferation of prostate cancer cells by inhibiting the FOXM1-mediated DNA damage response [44]. In breast cancer, NIC can effectively inhibit the STAT3 activation markers pY705 and pS727 and reduce the STAT3 dimerization capacity, thus significantly reducing the cell proliferation capacity [45].

#### 3.1.2. NIC Inhibits Tumor Cell Migration and Invasion

Another essential feature of tumor cells is the loss of contact inhibition and cell migration, which predispose tumors to metastasis and affect treatment prognosis and clinical survival. The migration and invasion of tumor cells mainly involve MAPK signaling [46], JAK–STAT signaling [47], Wnt signaling [48], TGF-β–Smad signaling, and PI3K/Akt-mTOR signaling [49]. NIC can inhibit the migration and invasion of tumor cells in various cancers, such as breast cancer, osteosarcoma, melanoma, liver cancer, glioma, oral squamous cell cancer, prostate cancer, and non-small-cell lung cancer [50]. In breast cancer, NIC reverses the adipocyte-induced epithelial–mesenchymal transition (EMT) in breast cancer cells by inhibiting the interleukin-6/STAT3 signaling axis, thus inhibiting cell migration and invasion ability [51]. In osteosarcoma, NIC inhibits the EMT by inhibiting the Wnt–Axin2–Snail cascade, thus inhibiting the migration and invasive capacity of a cell [52]. In melanoma, NIC inhibits p-STAT3 expression, thus inhibiting lung metastasis in melanoma [53]. In HCC, NIC inhibited cell migration and invasion ability by inhibiting CD10 expression in HCC cells [54]. In glioma, NIC promotes the overexpression of ALK4, which significantly downregulates the phosphorylation of JAK 2 and STAT3, thus inhibiting the migration and invasion ability of glioma cells [55]. In oral squamous cell carcinoma, NIC reduces its migration and invasion by inhibiting the let-7a/STAT3 axis [56]. Activation of the androgen receptor (AR) and its splice variants is linked to advanced prostate cancer and drives resistance to antiandrogens [57]. In prostate cancer, NIC inhibits the migration and invasion of resistant prostate cancer cells by inhibiting the IL6–STAT3–AR axis [58]. In non-small-cell lung cancer, NIC blocks S100A4 expression and function by inhibiting the NF-κB-mediated expression of MMP 9, thereby inhibiting cell migration and invasion [59].

#### 3.1.3. NIC Promotes Tumor Cell Apoptosis

Anti-apoptosis is also one of the important characteristics of tumor cells. The mechanisms of apoptosis mainly involve the mitochondrial pathway, the endoplasmic reticulum (ER) pathway, and the death receptor pathway. Studies showed that NIC promotes cell apoptosis of some cancers, such as leukemia, CRC, lung cancer, liver cancer, melanoma, prostate cancer, human chondrosarcoma, esophageal cancer, human thyroid cancer, and pancreatic cancer [60].

In CML, NIC induces apoptosis of CML cells by disabling Sp1 [39]. In CRC, NIC blocks DCLK1-B transcription by disrupting the binding of LEF1 to the DCLK1-B promoter, and reduces the expression of DCLK1-B, leading to increased apoptosis, thus making CRC more sensitive to radiotherapy and chemotherapy [42]. In lung cancer, NIC activates caspases to induce apoptosis through the death receptor pathway, while NIC elevates ROS levels via ER stress and mitochondrial potential loss [61]. In CRC, NIC promotes apoptosis by downregulating the expression of anti-apoptotic proteins Mcl-1 and Survivin, inhibiting the Notch signaling pathway, and upregulating the expression of miR-200 family members [62]. In melanoma, NIC induces energy stress, regulates the AMPK–mTOR pathway, and promotes apoptosis without affecting the MEK–ERK–MAPK signaling pathway [63]. In prostate cancer, NIC induces apoptosis by inhibiting the FOXM1-mediated DNA damage response [44]. In human chondrosarcoma, NIC activates the caspase-dependent mitochondrial apoptotic pathway [64]. In esophageal cancer, NIC promotes apoptosis by inhibiting Wnt/β-catenin signaling [65]. In human thyroid cancer, NIC activates Bax and caspase-3, and suppresses Bcl-2 and mitochondrial membrane potential (ΔYm), suggesting that NIC may induce apoptosis through the mitochondria-mediated endogenous apoptotic pathway [66]. In pancreatic cancer, NIC promotes mTORC1-dependent autophagy and cell death by targeting pGSK3β-mediated non-canonical Hedgehog signaling, thereby promoting apoptosis [67].

#### 3.1.4. NIC Regulates Cancer Cell Stemness

Elevated mitochondrial biogenesis and/or metabolism are distinguishing features of cancer cells, as well as cancer stem cells (CSCs), which are involved in tumor initiation, metastatic dissemination, and therapy resistance. In fact, mitochondria-impairing agents can be used to hamper CSC maintenance and propagation, which allows for better control of the neoplastic disease [68,69]. Combined treatment with NIC and an inhibitor of oxidative phosphorylation termed dodecyl-TPP (d-TPP) can reduce the proliferation of tumor stem cells [70]. In a zebrafish model that enabled 3D visualization of tumor cell extravasation, NIC significantly reduced tumor cell extravasation through the modulation of signaling pathways, chemokines, and tumor–endothelial cell interactions [71]. NIC exhibits high activity against BHGc7 tumorospheres (TOS) and UHGc5 TOS but not against the other circulating tumor cell (CTC) spheroids [72]. NIC with pronounced multicellular tumor spheroid (MCTS)-selective activity can inhibit mitochondrial respiration. This suggests that cancer cells in low glucose concentrations depend on oxidative phosphorylation rather than solely glycolysis [73]. Drug screening identified NIC as an inhibitor of breast cancer stem-like cells. NIC downregulates stem pathways, inhibits the formation of spheroids, and induces apoptosis in breast cancer side population spheres (SPS). Animal studies also confirmed this therapeutic effect [74].

#### 3.1.5. NIC Sensitizes Tumor Cells to Chemotherapy and Immunotherapy

Chemotherapy is the main means of tumor treatment that can reduce or eliminate tumors. However, two main factors restrict the success rate of chemotherapy, one is the toxic side effects of anti-tumor drugs, while the other is that tumor cells are resistant to anti-cancer drugs, which is the main cause of chemotherapy failure and patient death. Therefore, cancer treatment has shifted from the initial single-drug therapy to a combination drug therapy, which allows for complementary mechanisms, synergistic effects, and alleviation of adverse reactions [75]. The most important cause of tumor drug resistance is the overexpression of the “ABC transport pump” on the cell membrane in tumor stem cells [76]. As a model of “new use of old drugs”, NIC stands out among the many drug candidates against various resistance mechanisms [28]. Studies found that NIC has anti-tumor activity against sensitive and multidrug-resistant (P-glycoprotein overexpression) leukemia cells, which may be due to its rapid absorption and effective bypass of P-gP, resulting in higher intracellular accumulation and effectiveness [77]. At present, many reports in the literature demonstrate the improved drug sensitivity and efficacy of combinations of NIC with a variety of broad-spectrum anti-cancer drugs in the treatment of various cancers [78].

In all clinical subtypes of breast cancer, NIC combined with doxorubicin can jointly promote the death of all breast cancer cells by inhibiting the ROS signaling pathway and blocking the Wnt/β-catenin and G0/G1 cell cycle at different combined concentrations, thereby inducing apoptosis [79]. Especially in triple-negative breast cancer, the combination of NIC and paclitaxel showed a good inhibitory effect on breast cancer stem cells [80]. In cisplatin-resistant HER2-positive breast cancer, NIC combined with cisplatin can inhibit breast cancer cell invasion, the epithelial–mesenchymal transformation, and stem cell differentiation, suggesting that NIC combined with cisplatin may be a new treatment for HER2-positive breast cancer [81]. In APC-mutant CRC, NIC effectively inhibits Wnt signaling as well as Hippo signaling, which limits the therapeutic potential for CRC. To overcome this limitation, Kang He et al. used a combination of metformin with NIC, which not only inhibits canonical Wnt signaling activity but also inhibits YAP activity in CRC cells and patient-derived tumor organelles by inhibiting tumor stem cells, providing a new approach for clinical treatment of CRC [82]. Flavopiridol, which is an inhibitor of CDKs (cyclin-dependant kinases), is currently undergoing clinical trials for leukemia and other blood cancers; however, it has a strong cytotoxic effect on the skin. To combat this, X. H. Zhang et al. found that NIC can be used in combination with flavopiridol to prevent clinical adverse reactions [83]. NIC was identified as a combined drug candidate against ARA-C acute myeloid leukemia [83]. NIC induces apoptosis in castration-resistant prostate cancer and reduces the growth of xenograft tumors by inhibiting the FOXM1-mediated DNA damage response [44]. NIC is a novel inhibitor of AR-V7, where AR-V7 is related to the drug resistance of bicalutamide [84], enzalutamide [85], and abiraterone [86]. In addition, there are some clinical studies on the treatment of prostate cancer by NIC. Oral NIC could not be escalated above 500 mg TID (ter in die), and plasma concentrations were not consistently above the threshold shown to inhibit growth in castration-resistant prostate cancer (CRPC) models, which means oral NIC is not a viable compound for repurposing as a CRPC treatment [87]. A phase Ib trial of reformulated NIC with abiraterone/prednisone in men with castration-resistant prostate cancer (CRPC) showed that NIC/PDMX1001 reformulation achieved targeted plasma levels when combined with abiraterone and prednisone, and it was well tolerated, which means the further study of NIC/PDMX1001 with this combination is warranted [88]. 

Immunotherapy is a therapeutic strategy that mobilizes the activity of immune cells in the body, reactivates the inactivated immune cells, and indirectly kills tumor cells. So far, it has shown strong anti-tumor activity in the treatment of a variety of tumors, such as melanoma, non-small-cell lung cancer, kidney cancer, and prostate cancer. However, there are still some limitations. For example, some patients do not respond to immunotherapy due to the immunosuppressive mechanism of the tumor microenvironment, low effect, and the cause of autoimmune disorders [89]. NIC can inhibit cancer progression by modulating immune pathways. NIC enhances the PD-L1 antibody in the inhibition of non-small-cell lung cancer (NSCLC) growth in vitro and in vivo, which was involved in the blockage of p-STAT3 binding to the promoter of PD-L1 and finally downregulation of PD-L1 expression. These encourage the combination therapy of NIC and PD-1/PD-L1 blockade to be further studied in the clinic [90].

**Table 1 ijms-23-16116-t001:** The pharmacological activity and mechanism of NIC and its derivates in cancer.

Name	Mechanism	References
Leukemia	Inhibition of proliferation by inhibiting the Wnt/β-catenin signaling pathway and downregulating phosphorylated CREB, STAT5, and Akt expression. Induction of apoptosis by disabling Sp1.	[38,39]
Nasopharyngeal carcinoma	Inhibition of proliferation by downregulating ku70/80 expression.	[40]
Hepatoma	Inhibition of proliferation by inhibiting the STAT3 signaling pathway.	[41]
Colorectal cancer	Inhibition of proliferation by downregulating DCLK1 expression. Induction of apoptosis by reducing DCLK1-B, Mcl-1, and survivin expression; inhibiting the Notch signaling pathway; and upregulating miR-200 family members’ expression. Prevention of chemotherapeutic resistance by inhibiting Wnt, Hippo, and YAP.	[42,62,82]
Colon cancer	Inhibition of proliferation by promoting mitochondrial decoupling.	[16,25,43]
Prostate cancer	Inhibition of proliferation and induction of apoptosis by inhibiting the FOXM1-mediated DNA damage response. Inhibition of migration and invasion by inhibiting the IL6–STAT3–AR axis.	[44,58]
Breast cancer	Inhibition of proliferation by inhibiting STAT3 activation markers pY705 and pS727 and reducing the STAT3 dimerization capacity. Inhibition of migration and invasion by inhibiting the interleukin-6/STAT3 signaling axis. Prevention of chemotherapeutic resistance by targeting ROS and Wnt/β-catenin.	[45,51,79]
Osteosarcoma	Inhibition of migration and invasion by inhibiting the Wnt–Axin2–Snail cascade.	[52]
Melanoma	Inhibition of migration and invasion by inhibiting p-STAT3 expression. Induction of apoptosis by regulating the AMPK–mTOR pathway.	[53,63]
Hepatocellular carcinoma	Inhibition of migration and invasion by inhibiting CD10 expression.	[54]
Glioma	Inhibition of migration and invasion by overexpressing ALK4 (NIC).	[55]
Oral squamous cell carcinoma	Inhibition of migration and invasion by inhibiting the let-7a/STAT3 axis.	[56]
Lung cancer	Inhibition of migration and invasion by blocking S100A4 expression. Induction of apoptosis by activating caspases. Enhances immunotherapy efficiency by enhancing PD-L1 antibodies.	[59,61,90]
Chondrosarcoma	Induction of apoptosis by activating the caspase-dependent mitochondrial apoptotic pathway.	[64]
Esophageal cancer	Induction of apoptosis by inhibiting the Wnt/β–catenin signaling pathway.	[65]
Thyroid cancer	Induction of apoptosis by activating Bax and caspase-3 and suppressing Bcl-2 and mitochondrial membrane potential (ΔYm).	[66]
Pancreatic cancer	Induction of apoptosis by targeting the p-GSK3β-mediated non-canonical Hedgehog signaling pathway.	[67]

#### 3.1.6. NIC’s Formulations for Treating Cancer

Now there are some new formulations to improve the water solubility of NIC and reduce its toxicity, leading to better tumor treatment effects. An injectable pegylated NIC (polyethylene glycol-modified NIC) was synthesized. The water solubility of NIC in mPEG5000-Nic was significantly increased. Meanwhile, mPEG5000-Nic was less toxic, which indicated that pegylated NIC is a novel NIC delivery system with clinical potential for cancer therapy [91]. It was found that the solubility and dissolution of NIC can be improved by using octenylsuccinate hydroxypropyl phytoglycogen (OHPP), which is an amphiphilic dendrimer-like biopolymer. It also has a stronger inhibitory effect on cancer cell lines [92]. Nanoliposomal encapsulation enhances the aqueous solubility of NIC and improves its anti-tumor properties [93]. NIC conjugates to recombinant chimeric polypeptides (CPs), and the CP-NIC conjugate spontaneously self-assembles into sub-100 nm near-monodisperse nanoparticles. CP-NIC nanoparticles delivered intravenously act as a pro-drug of NIC to dramatically increase the exposure to NIC compared with dosing with free NIC [94]. Co-crystals of NIC-nicotinamide (NIC-NCT) have improved solubility characteristics (≥14.8-fold) relative to the pure drug. NIC-NCT showed a superior cytotoxic activity compared with the pure drug. Mechanistically, NIC-NCT co-crystals enhanced the autophagic flux in cancer cells, which demonstrates autophagy-mediated cell death [95]. An NIC stearate prodrug therapeutic (NSPT) formulation stabilizes NIC stearate against hydrolysis and delays enzymolysis; increases circulation in vivo with t_1/2_ approximately 5 h; reduces cell viability and cell proliferation in human and canine osteosarcoma cells in vitro at 0.2–2 μmol/L IC_50_; inhibits recognized growth pathways and induces apoptosis at 20 μmol/L; eliminates metastatic lesions in an ex vivo lung metastatic model; and when injected intravenously at 50 mg/kg weekly, it prevents metastatic spread in the lungs in a mouse model of osteosarcoma over 30 days [96]. NIC nanocrystals present a comparable anti-tumor effect to the drug solution against an EC9076 cell line. Therefore, a nanocrystal formulation with solution-like behaviors should be a promising choice for the intravenous delivery of NIC [97]. An inclusion complex of NIC with cyclodextrin was prepared using a freeze-drying method; in in vitro cytotoxicity studies, this complex indicated significantly higher cytotoxicity at lower concentrations, while pharmacokinetic studies showed significant improvement in the C_max_ and T_max_ of NIC from cyclodextrin complex in comparison to pure NIC alone [98]. A nanosuspension of NIC (nano-NIC) showed rapid absorption (reaching the maximum plasma concentration within 5 min) and improved the bioavailability (the estimated bioavailability for oral nano-NI was 25%). In conclusion, nano-NI has the potential to be a new treatment modality for ovarian cancer [99]. NIC nanocrystals (NLM-NCs) have higher solubility and storage stability. NLM-NCs can inhibit cell migration, as well as decrease the expression of CD44, which is a marker of breast cancer stem cells [100]. 

### 3.2. The Metabolic Regulatory Activity of NIC

At present, there are many reports in the literature on the metabolic regulatory activity of NIC, mainly involving the regulation of metabolic diseases or metabolic disorders, including obesity, diabetes, and non-alcoholic fatty liver disease (Table 2). NEN can inhibit the progression of type 1 diabetic nephropathy by reducing urinary albumin levels and improving renal hypertrophy, which reduces podocyte dysfunction NEN can increase lipid metabolism, uncouple kidney mitochondria, and significantly inhibit renal cortical activation of the mTOR/4E-BP1 pathway [19]. In addition, it has a protective effect on the liver and is not cardiotoxic. Therefore, these findings open up a whole new therapeutic approach for diabetes and diabetic kidney disease [15]. At the same time, the drugs currently used to treat type 2 diabetes ameliorate the hyperglycemic symptoms of the disease, but the underlying mechanism of hyperglycemia has not been fully resolved. Inhibition of glucagon signaling contributes to the beneficial effects of NIC and NEN on systemic glucose metabolism. The results suggest that NIC may be a useful adjunctive therapeutic strategy for type 2 diabetes [101]. In addition, NIC was reported to improve non-alcoholic fatty liver disease by inducing the AMPK-mediated phosphorylation of p62 at S351 to cause atypical Nrf2 activation. NIC also protects the liver from acute lipotoxic stress by activating the p62-dependent Keap1–Nrf2 pathway [25]. 

### 3.3. The Immune Disease Therapeutic Activity of NIC

Many studies showed that NIC has activity in the treatment of immune diseases. The main diseases in which NIC has therapeutic activity are rheumatoid arthritis [102], graft-versus-host disease [103], systemic sclerosis [104], and systemic lupus erythematosus, among other diseases [17] (Table 2).

It was reported that NIC has a therapeutic effect on rheumatoid arthritis by reducing TNF-α-induced cytokine expression and release in synovial cells of human fibroids with rheumatoid arthritis [105]. In addition, NIC inhibits serum-induced synovial cell migration and invasion and causes changes in the cellular filamentous actin cytoskeleton network by decreasing TNF-α-stimulated MAP kinase production and IKK/NF-κB signaling activity in synovial cells [106]. Studies also showed that NIC significantly alleviates the degree of injury in collagen-induced arthritis rat models [107]. Meanwhile, NIC can induce the apoptosis of human rheumatoid arthritis fibroblast synovial cells [108]. In scleroderma graft-versus-host disease (GVHD) models, NIC was reported to reverse the clinical symptoms of GVHD, including alopecia, vasculitis, and diarrhea [103], of which beneficial effects were associated with the inhibition of the STAT3, Wnt/β-catenin, ERK1/2, AKT, and Notch signaling pathways. It was found that NEN alleviated systemic lupus erythematosus and lupus nephritis in mice and reduced urinary protein excretion. In addition, a diet supplemented with NEN can restore redox imbalance, promote mitochondrial production, and improve renal energy imbalance. More importantly, NEN prevented Swollen lymph nodes and splenomegaly and reduced serum anti-dsDNA antibody levels in mice. Therefore, NIC and its derivate NEN is a great potential drug for autoimmune diseases [17].

### 3.4. The Anti-Infective Activity of NIC

#### 3.4.1. The Antiviral Activity of NIC

NIC has broad-spectrum antiviral activities against, for example, severe acute respiratory syndrome coronavirus (SARS) [6], Middle East respiratory syndrome coronavirus (MERS-CoV) [109], Zika virus (ZIKV) [110], Japanese encephalitis virus (JEV) [111], hepatitis C virus (HCV) [6], Ebola virus (EBOV) [112], human rhinovirus (HRV) [113], Chikungunya virus (CHIKV) [114], human adenovirus (HADV) [115], and Epstein–Barr virus (EBV) [116] (Table 3). It was reported that NIC can inhibit the replication and cytopathic effect (CPE) of SARS coronavirus at low concentrations of 1 M and eliminate viral antigen synthesis at 1.56 M [117]. Studies showed that NIC can inhibit MERS-CoV replication at 1000-fold, but the specific mechanism has not been clarified [118]. Meanwhile, NIC was also found to inhibit ZIKV replication in brain tissue [119]. The combination of NIC with PF-03491390, which is a nonselective caspase inhibitor, further enhanced the protective effect of human neural progenitors and astrocytes against ZIKV-induced cell death [120]. Some studies identified NIC as a potent JEV inhibitor with a micromolar titer. Time-dependent experiments showed that NIC inhibited the proliferation of the JEV in the replication phase [111]. NIC provides good anti-HCV replication activity by inhibiting the replication of HCV host cells [121]. NIC, which is a weak lipophilic acid, was reported to inhibit PH-dependent HRV infection at micromolar concentrations; the main mechanism involved acting as a proton carrier to inhibit the entry of HRV by blocking the acidification of endolysosomal compartments [113]. NIC was found to be an effective anti-CHIKV inhibitor by blocking the entry of low-pH-dependent CHIKV [114]. By inhibiting the transport of HADV particles from the endosome to the nuclear membrane, NIC shows good anti-HADV activity at a low micromolar value [122]. In addition, NIC also inhibits the division and replication of EBV in cells and offers potential for the treatment of acute EBV-associated infectious diseases by interfering with the irreversible cell cycle arrest of mTOR-activated mitotic EBV-infected cells [116]. Therefore, NIC is regarded as a low-cost drug with extensive antiviral properties that show extremely promising potential for clinical development.

#### 3.4.2. The Antibacterial Activity of NIC

NIC also showed therapeutic potential against bacterial diseases, such as *mycobacteria*, *M. tuberculosis*, *Bacillus anthracis*, *Pseudomonas aeruginosa*, and *Staphylococcus aureus* (Table 3). The inhibitory effect of NIC on the growth of *M. tuberculosis* is pHdependent, and the mechanism acts by affecting the cell-mediated immune response [123]. NIC protects cells exposed to macrophages from anthrax toxin or cells from *Pseudomonas aeruginosa* exotoxin and diphtheria toxin [124]. NIC has an anti-*Staphylococcus*-*aureus* infection effect, but the mechanism has not been defined [125]. Therefore, NIC has broad antibacterial activity.

### 3.5. The Other Pharmacological Activities of NIC

In addition to the therapeutic role in the above-described diseases, NIC also has inhibitory effects against smooth muscle contraction, mainly in the treatment of asthma and the inhibition of arterial contraction. NEN can relieve the contraction of the mesenteric artery induced by phenylephrine (PE) and high K^+^ (KPSS) [126] (Table 2), where the inhibitory effect of NEN on arterial constriction suggests that it has broad application prospects as an antihypertensive drug, but it may also cause vasodilation-related side effects after absorption in vivo [14]. The reposition of NEN can be used as a treatment for asthma, and its main mechanism is to inhibit bronchial smooth muscle contraction by activating the AMPK pathway and suppress bronchial smooth muscle cell proliferation and migration by inhibiting the STAT3 pathway [21]. Recently, Zhen Liu et al. found that NIC modulates the development of myopia driven by canonical Wnt signaling, where a mouse model treated with NIC showed significant inhibition of Wnt signaling and reduced lens thickness, vitreous cavity depth, and axial length, thereby inhibiting myopia [127].

**Table 2 ijms-23-16116-t002:** The pharmacological activities and mechanisms of NIC and its derivates in metabolic syndrome, the immune system, and other types of diseases.

Diseases		Mechanism	References
Metabolic Syndrome	Diabetes	Downregulation of the mTOR/4E-BP1 signaling pathway in type 1 diabetes. Inhibition of glucagon signaling in type 2 diabetes.	[15,19,101]
	Nonalcoholic steatohepatitis	Induction of the AMPK-mediated phosphorylation of p62 (S351).	[25]
	Non-alcoholic fatty liver disease	Induction of the AMPK-mediated phosphorylation of p62 at S351 to cause atypical Nrf2 activation.	[25]
	Acute lipotoxic stress	Induction of the p62-dependent Keap1–Nrf2 signaling pathway.	[25]
Immune system diseases	Rheumatoid arthritis	Induction of apoptosis by reducing TNF-α-induced cytokine expression, MAP kinase production, and the IKK/NF-κB signaling activity.	[105,106,107,108]
	Graft-versus-host disease	Inhibition of the STAT3, Wnt/β-catenin, ERK1/2, AKT, and Notch signaling pathway.	[103]
	Systemic lupus erythematosus and lupus nephritis	Reduction of urinary protein excretion. Restoring the redox imbalance. Promotion of mitochondrial production. Improvement of renal energy imbalance.	[17]
	Swollen lymph nodesand splenomegaly	Reduction of serum anti-dsDNA antibody levels.	[17]
Other types	Athma	Activation of the AMPK pathway. Inhibition of bronchial smooth muscle cell proliferation and migration by inhibiting the STAT3 pathway.	[21]
	Arterial constriction	Effect on smooth muscle contraction.	[14,126]
	Myopia	Inhibition of the Wnt signaling pathway.	[127]

**Table 3 ijms-23-16116-t003:** The pharmacological activities and mechanisms of NIC and its derivates against virus and bacterial infection.

Diseases		Mechanism	References
Virus	Coronavirus	Inhibition of MERS-CoV and SARS-CoV replication and viral antigen synthesis.	[117,118]
	Zika virus	Inhibition of Zika virus replication.	[119,120]
	Japanese encephalitis virus	Inhibition of proliferation.	[111]
	Hepatitis C virus	Inhibition of the replication of HCV host cells.	[121]
	Human rhinovirus	Blocks the acidification of the endolysosomal compartment.	[113]
	Chikungunya virus	Block the entry of low-pH-dependent CHIKV.	[114]
	Human adenovirus	Inhibition of the transport of HADV particles from the endosome to the nuclear membrane.	[122]
	Epstein–Barr virus	Inhibition of irreversible cell cycle arrest activated by mTOR.	[116]
Bacteria	*Tuberculosis*	Effects on the cell-mediated immune response.	[123]
	*Pseudomonas aeruginosa*	Not clear	[124]
	*Staphylococcus aureus*	Not clear	[125]

## 4. Conclusions and Future Perspectives

In the past, NIC was used as an oral antihelminth drug to treat parasitic infections [27], while it was shown therapeutic potential in a variety of human diseases and cancer in recent years. NIC has long been approved by the FDA due to its low cost, low cytotoxicity, and high water solubility, which make it a rising star in the “old medicine” category and sought after by companies and researchers [15]. Now NIC has emerged in cancer, metabolic diseases, immune system diseases, virus and bacteria diseases, and other diseases. With the deepening of research, NIC not only plays a role in mitochondrial uncoupling in various diseases but also regulates cell proliferation, migration, invasion, and apoptosis through Wnt/β-catenin, mTORC1, STAT3, NF-KB, Notch, and other signaling pathways [30,31,32,33,34]. For cancers with no specific targeted therapy or drug resistance, combining NIC with other drugs or immunotherapy can also achieve better therapeutic effects. However, NIC is not perfect. It has high cytotoxicity and low water solubility, which limits its wide application as an oral drug [15]. Therefore, some researchers have started to pay attention to the derivatives of NIC. NIC ethanolamine salt (NEN) and NIC piperazine (NPP) are the two main salt forms of NIC that were widely studied. They have similar effects to NIC and have higher water solubility and safety [13]. In particular, NEN has higher water solubility and higher safety, and the shortage of NIC was greatly improved, which provides a better prospect for clinical application. Meanwhile, several studies were carried out to improve the delivery methods of NIC to eliminate its cytotoxicity [128], which will truly make it possible for NIC to be applied to the clinical treatment of human disease and cancer.

## 5. Outstanding Questions

Despite the therapeutic activity of NIC in a variety of human diseases, it still has limitations, such as cytotoxicity and low water solubility, which limit its clinical application. Therefore, studies based on overcoming the limitations of NIC are crucial. Research showed that the toxicity of NIC in the human body may be related to NIC being subjected to efficient metabolic reactions, namely, hydroxylation and glucuronidation, wherein CYP1A2 and UGT1A1 were the main contributing enzymes, respectively [129]. This gives us a direction regarding the mechanism of the toxicity of NIC in the human body. To further enhance the therapeutic effects of NIC, it is necessary to study the mechanism of toxicity of NIC, modify its derivatives, develop a drug combination strategy, or develop new delivery methods to further overcome the limitations of NIC in the future.

## Figures and Tables

**Figure 1 ijms-23-16116-f001:**
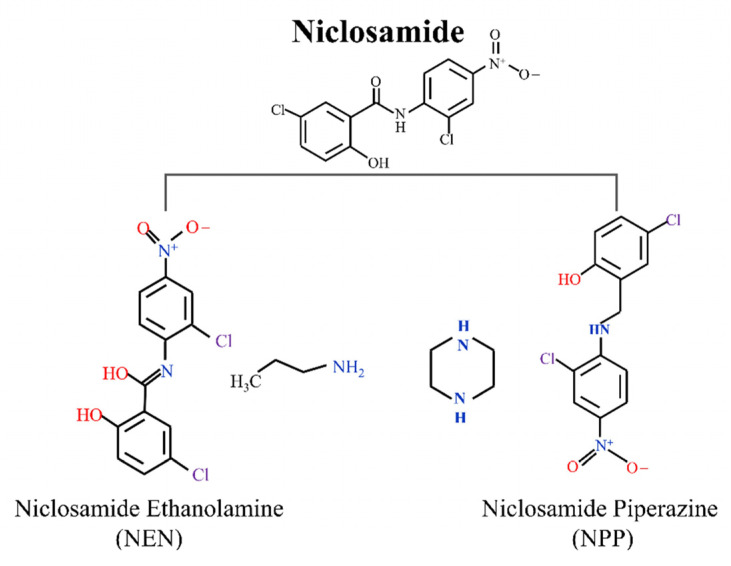
The structure of the two main salt forms of NIC: NEN and NPP.

**Figure 2 ijms-23-16116-f002:**
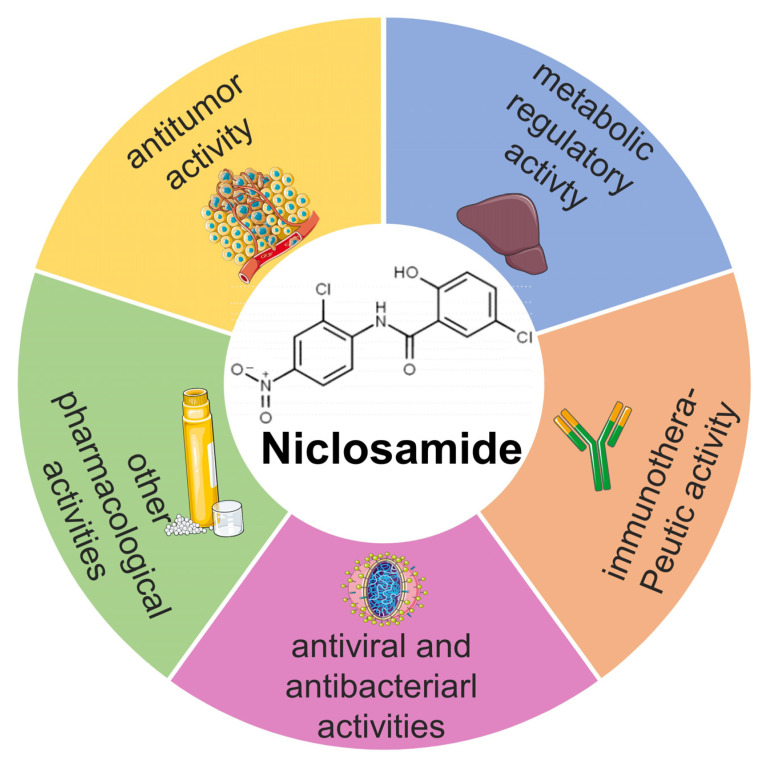
The multifaceted pharmacological activities of NIC in human disease and cancer.

**Figure 3 ijms-23-16116-f003:**
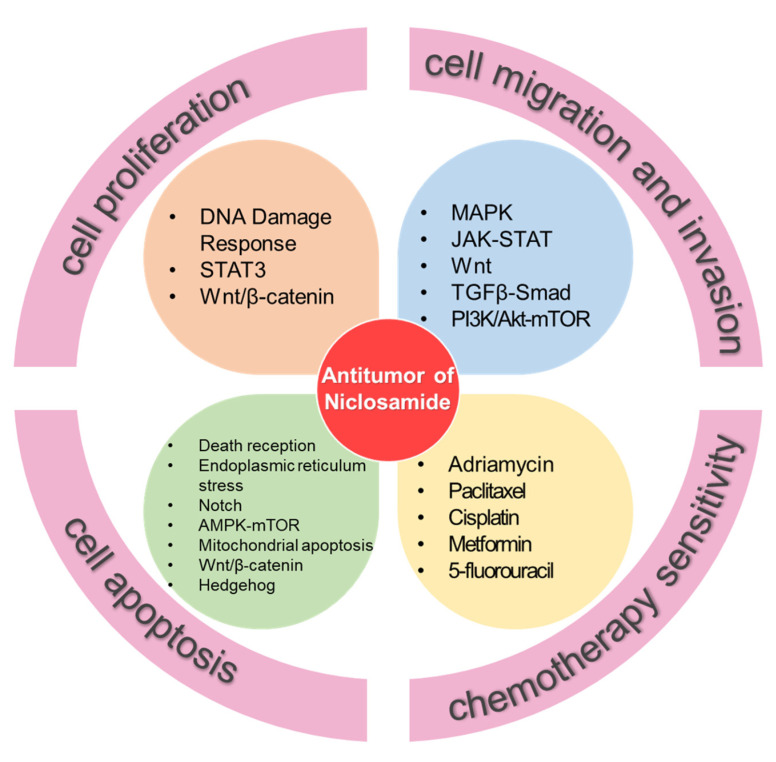
The anti-tumor activity and mechanism of NIC.

## Data Availability

All data generated and/or analyzed during the current study are available.

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
