# Peer review of "Niclosamide as a Promising Therapeutic Player in Human Cancer and Other Diseases"

_ijms, 2022, doi:10.3390/ijms232416116_

Round 1
Reviewer 1 Report
Present paper dealing with the interesting topic. The article can be considered for publication as a summary of the recent works is correct. The paper can be published as it is, with some minor corrections.
- The structure of ethanolamine salt should be presented as the author mentioned about it.
- Due to too many abbreviations, abbreviations should be listed before the References section.
Author Response
The point-by-point answers to the Editors and the reviewers are listed below.
Dear Editors and Reviewers,
We appreciate the editors and reviewers for their constructive criticisms that have helped us to improve the revised manuscript, entitled “Niclosamide as a promising therapeutic player in human disease and cancer
” (ID: ijms-2019223). According to your valuable advice, we have made careful corrections which we hope to meet with approval. We performed a scientific English editing of our revised manuscript and modified the authors’ emails.
For your reading convenience, the revised portions are marked in red in the manuscript, and the responses to the reviewers’ comments are as follows:
Sincerely,
Guiling Wang
Point-by-point responses to the comments of the expert reviewers
For your reading convenience, the comments from the expert reviewers are reproduced in italics and our point-by-point responses are shown in dark blue.
Reviewers’ Comments:
Referee #1
Major points:
Present paper dealing with the interesting topic. The article can be considered for publication as a summary of the recent works is correct. The paper can be published as it is, with some minor corrections.
The structure of ethanolamine salt should be presented as the author mentioned about it.
Response: Thank you for your valuable suggestions, which make the description of our manuscript more systematic and complete. We have added the description and structural figure (shown in Figure 1) of two salt derivatives of niclosamide in the revised manuscript. (Page 1 Line 45 to Page 3 Line 89)
Due to too many abbreviations, abbreviations should be listed before the References section.
Response: We agree with the comment and supplied abbreviations before the References section in the revised manuscript. (Page 12 Line 452 to Page 13 Line 467)
Reviewer 2 Report
In this manuscript titled "Niclosamide as a promising therapeutic player in human disease and cancer" Wang et al reviewed the pharmaceutical activities and therapeutic prospects of niclosamide in human disease and cancer and summarized the related molecular mechanisms and signaling pathways, indicating niclosamide is a promising therapeutic player in human various diseases including cancer.
Although the manuscript is well-written, I have some suggestions to make hereby as follows:
1) Please check the author line and their respective affiliations.
2) For section 2.2. it would be better if authors can included a mechanistic pathway.
3) Please arrange the reference according to the journal guidelines.
This can be accepted after including the above minor revisions.
Author Response
The point-by-point answers to the Editors and the reviewers are listed below.
Dear Editors and Reviewers,
We appreciate the editors and reviewers for their constructive criticisms that have helped us to improve the revised manuscript, entitled “Niclosamide as a promising therapeutic player in human disease and cancer
” (ID: ijms-2019223). According to your valuable advice, we have made careful corrections which we hope to meet with approval. We performed a scientific English editing of our revised manuscript and modified the authors’ emails.
For your reading convenience, the revised portions are marked in red in the manuscript, and the responses to the reviewers’ comments are as follows:
Sincerely,
Guiling Wang
Point-by-point responses to the comments of the expert reviewers
For your reading convenience, the comments from the expert reviewers are reproduced in italics and our point-by-point responses are shown in dark blue.
Reviewers’ Comments:
Reviewer #2
Major points:
In this manuscript titled "Niclosamide as a promising therapeutic player in human disease and cancer" Wang et al reviewed the pharmaceutical activities and therapeutic prospects of niclosamide in human disease and cancer and summarized the related molecular mechanisms and signaling pathways, indicating niclosamide is a promising therapeutic player in human various diseases including cancer.
Although the manuscript is well-written, I have some suggestions to make hereby as follows:
- Please check the author line and their respective affiliations.
Response: We apologize for the problems in the original manuscript. We have adjusted the format here of authors and our affiliations in the revised manuscript. . (Page 1 Line 4 to Page 1 Line 8)
- For section 2.2. it would be better if authors can included a mechanistic pathway.
Response: We are grateful for the suggestion. As suggested by the reviewer, we have added more details of the mechanistic pathway in the revised manuscript. (Page 9 Line 313 to Page 9 Line 315, Page 9 Line 325)
- Please arrange the reference according to the journal guidelines.
Response: Thank you for your precious comments and advice. We have revised the format of references according to the journal guidelines in the revised manuscript.
This can be accepted after including the above minor revisions.
Reviewer 3 Report
This article provides a very boring and routine account of niclosamide's biological applications, which is very superficial and adds nothing of value.
Author Response
The point-by-point answers to the Editors and the reviewers are listed below.
Dear Editors and Reviewers,
We appreciate the editors and reviewers for their constructive criticisms that have helped us to improve the revised manuscript, entitled “Niclosamide as a promising therapeutic player in human disease and cancer
” (ID: ijms-2019223). According to your valuable advice, we have made careful corrections which we hope to meet with approval. We performed a scientific English editing of our revised manuscript and modified the authors’ emails.
For your reading convenience, the revised portions are marked in red in the manuscript, and the responses to the reviewers’ comments are as follows:
Sincerely,
Guiling Wang
Point-by-point responses to the comments of the expert reviewers
For your reading convenience, the comments from the expert reviewers are reproduced in italics and our point-by-point responses are shown in dark blue.
Reviewers’ Comments:
Reviewer #3
Major points:
This article provides a very boring and routine account of niclosamide's biological applications, which is very superficial and adds nothing of value.
Response: Thank you for your review. Our manuscript does have some problems. In this revision, we have added a description of niclosamide’s derivatives, as well as a description of the pharmacokinetics of niclosamide and the new formulations for treating cancer, hoping to satisfy you.
Reviewer 4 Report
The manuscript is a review of the publications dedicated to the potential applications of an old anthelmintic drug, niclosamide in human diseases. However, it does not provide too much new knowledge as several similar reviews exist in literature. Thus, the addition of new data would increase the quality and novelty of the review. For example, the Authors may add information about pharmacokinetic properties, including metabolic pathways, of niclosamide and its salt. Moreover, more derivatives of this drug may be presented together with their physiochemical and pharmacokinetic properties and pharmacological activity.
A section about new formulations targeting cancer of niclosamide may be included.
The title suggests that cancer is not a human disease.
The Authors have not mentioned that the drug is no longer commercially available in the United States (line 26).
The abbreviation NIC used in Table 1 has not been introduced in the text.
The expression “pharmaceutical activity “ (Fig. 1) is not clear. Should be rather “pharmacological”.
Tables 1-3- each sentence starts with the word NIC or NEN. They should rather start with “inhibition of”, “induction of”, etc.
Line 169 – it should be clarified whether niclosamide is a P-gp substrate.
Line 307 - what does “remodeling of NEN” exactly mean?
Lines 311-312 - ”mouse model treated with niclosamide treated mouse models” is unclear
Author Response
The point-by-point answers to the Editors and the reviewers are listed below.
Dear Editors and Reviewers,
We appreciate the editors and reviewers for their constructive criticisms that have helped us to improve the revised manuscript, entitled “Niclosamide as a promising therapeutic player in human disease and cancer
” (ID: ijms-2019223). According to your valuable advice, we have made careful corrections which we hope to meet with approval. We performed a scientific English editing of our revised manuscript and modified the authors’ emails.
For your reading convenience, the revised portions are marked in red in the manuscript, and the responses to the reviewers’ comments are as follows:
Sincerely,
Guiling Wang
Point-by-point responses to the comments of the expert reviewers
For your reading convenience, the comments from the expert reviewers are reproduced in italics and our point-by-point responses are shown in dark blue.
Reviewers’ Comments:
Reviewer #4
Major points:
The manuscript is a review of the publications dedicated to the potential applications of an old anthelmintic drug, niclosamide in human diseases. However, it does not provide too much new knowledge as several similar reviews exist in literature. Thus, the addition of new data would increase the quality and novelty of the review. For example, the Authors may add information about pharmacokinetic properties, including metabolic pathways, of niclosamide and its salt. Moreover, more derivatives of this drug may be presented together with their physiochemical and pharmacokinetic properties and pharmacological activity.
Response: Thank you for underlining this deficiency. According to your suggestion, we had added information and structural figure (shown in Figure 1) in the revised manuscript. (Page 1 Line 45 to Page 3 Line 89)
A section about new formulations targeting cancer of niclosamide may be included.
Response: Thank you for your suggestions to make our manuscript more innovative and valuable. We added a section describing the new formulations of niclosamide for cancer treatment in the revised manuscript. (Page 8 Line 269 to Page 9 Line 307)
The title suggests that cancer is not a human disease.
Response: Thank you for your careful review. We have revised the title in the revised manuscript. (Page 1 Line 1 to Page 1 Line 2)
The Authors have not mentioned that the drug is no longer commercially available in the United States (line 26).
Response: Thank you for pointing out our shortcomings. We have supplemented this content that the drug is no longer commercially available in the United States in the revised manuscript. (Page 1 Line 27 to Page 1 Line 28)
The abbreviation NIC used in Table 1 has not been introduced in the text.
Response: Thank you for your careful review. We added the abbreviation NIC of niclosamide in the revised manuscript. (Page 1 Line 21)
The expression “pharmaceutical activity “ (Fig. 1) is not clear. Should be rather “pharmacological”.
Response: Your suggestions are very useful to us. Since we added a new figure in the revised manuscript as Figure 1, figure 1 in the original manuscript was named as figure 2. We have revised the description. (Page 4 Line 98)
Tables 1-3- each sentence starts with the word NIC or NEN. They should rather start with “inhibition of”, “induction of”, etc.
Response: We deeply appreciate the reviewer’s suggestions. We have modified the contents of tables 1-3, so that each sentence of tables 1 to 3 is unified as “inhibition of”, “induction of”, etc.
Line 169 – it should be clarified whether niclosamide is a P-gp substrate.
Response: We are extremely grateful to the reviewer for pointing out this problem. We have added a description of the relationship between niclosamide and P-gp in the revised manuscript. (Page 7 Line 218 to Page 7 Line 221)
Line 307 - what does “remodeling of NEN” exactly mean?
Response: Thank you for pointing out our inaccurate description. We have modified the unsuitable statement “remodeling of NEN” to “reposition of NEN” in the revised manuscript. (Page 11 Line 398)
Lines 311-312 - ”mouse model treated with niclosamide treated mouse models” is unclear
Response: Thank you for underlining this deficiency. We have corrected the wrong english expression in the revised manuscript. (Page 11 Line 402 to Page 11 Line 404)
Round 2
Reviewer 3 Report
I believe the review article has been significantly improved by the authors and can be accepted in its current form.
Reviewer 4 Report
The manuscript, especially new sections (in red), requires editing. There are many errors and unclear expressions, in the text, for example:
The mechanism in cancer of NEN of NEN in cancer mainly involves?
with known safet.?
New sentences starting with And: And the water (l. 272)
And it is found that (l. 275)
and desirs storage stability (l. 306)
3.5. The other pharmaceutical activities of niclosamide – rather pharmacological
3.1.6. formulations of treating cancer
The abbreviation NIC introduced in the first line should be used throughout the text.
